# Green light enhances the phytochemical preservation of lettuce during postharvest cold storage

Shafieh Salehinia[1], Fardad Didaran[1], Sasan Aliniaeifard[2‡], Saman Zohrabi[1], Sarah MacPherson[1‡], Mark Lefsrud[1] *

1 Department of Bioresource Engineering, Macdonald Campus, McGill University, Montreal, Quebec, Canada, 2 Controlled Environment Agriculture Center, College of Agriculture and Natural Resources, University of Tehran, Tehran, Iran

☯ These authors contributed equally to this work.
‡ SA and SM also contributed equally to this work.
* mark.lefsrud@mcgill.ca

**Data Availability Statement:** All data files are available from the DRYAD database (URL: https://datadryad.org/stash/share/ngsVhWA13Txrbll92ZsrdhFgkmpA7sqCmyLnHTSUflk).

## Abstract

The postharvest lighting environment is a main factor that influences quality preservation for harvested biomass. The objective of this study was to evaluate postharvest changes in bioactive compounds of lettuce with different storage light spectra. The effects of green LEDs with peaks at 500 nm and 530 nm, white LEDs (400–700 nm), and dark storage were evaluated, where light intensity (10 µmol m$^{-2}$ s$^{-1}$) and photoperiod (12 h per day) were constant with air temperature at 5°C over the 14 d treatment period. Lettuce stored with 500 nm and 530 nm green LEDs exhibited 1474.5% and 1451.8% (approximately 15.7 and 15.5 times) higher antioxidant activity, respectively, compared to dark storage. Significant improvements in total phenolic content, and 67.5% and 64.8% increases in total soluble solids with 530 nm and 500 nm green LEDs over dark storage were discerned. Exposure to 530 nm green LEDs led to 128.2% (approximately 2.28 times) higher anthocyanin content, a 26.2% increase in carotenoids, and a 95% rise in flavonoid content compared to dark storage. Increases of 26.4% and 16.0% in chlorophyll a content in lettuce stored under 500 nm and 530 nm green LEDs, respectively, and 65.6% and 46.6% rises in the Chlorophyll a/b ratio were observed. Compared to dark storage, green LEDs (500 nm) resulted in a 13.5% higher total chlorophyll content. Findings underscore the positive impact of green LEDs on the nutritional quality of lettuce, providing insight for postharvest practices.

## Introduction

Lettuce (*Lactuca sativa*) is a globally distributed and consumed leafy green, valued for its high water content, low-calorie count, and rich nutritional profile, including dietary fiber, minerals, vitamins, antioxidants, and phytochemicals such as carotenoids and phenolic compounds [1–4]. Maintaining lettuce quality postharvest presents substantial challenges due to its continued metabolic activity after harvest, leading to chlorophyll degradation, wilting, and decay [5–8].

**Funding:** This work was funded by UTechnology Corporation (Calgary, Alberta, Canada) and Natural Sciences and Engineering Research Council of Canada (NSERC #CRDPJ 524170-18). However, the funders had no role in study design, data collection and analysis, decision to publish, or preparation of the manuscript.

**Competing interests:** The authors have declared that no competing interests exist.

These changes not only diminish the vegetable's appearance and nutritional value but lead to consumer rejection [1, 5–7]. Improving postharvest storage conditions directly benefits the economic viability of lettuce by extending its shelf life, maintaining its nutritional and visual quality for longer periods, and reducing waste. This ensures that more of the harvested crop reaches consumers in optimal condition, thereby maximizing market value and profitability.

Various methods have been employed to preserve postharvest quality, including chemical treatments, edible coatings, and modified atmosphere packaging [9–13]. While these techniques can be effective, they often come with concerns related to cost, effectiveness, practicality, and food safety [14, 15]. There is a growing interest in developing alternative methods that are both cost-effective and environmentally friendly. Among these, light-emitting diodes (LEDs) have emerged as a promising technology [16–19]. LEDs offer advantages such as precise control over light intensity and spectrum, non-toxicity, and cost efficiency [20–22].

LEDs can positively influence the nutritional attributes of various fruits and vegetables [23–26], with most research focusing on red and blue light. However, limited studies have explored the effects of green LEDs on bioactive compounds such as flavonoids and anthocyanins, particularly in leafy greens. Green LED light has been shown to influence various physiological and biochemical processes in plants, primarily through its interaction with specific photoreceptors like cryptochromes and phytochromes, which regulate numerous metabolic pathways. Unlike red and blue light, which predominantly activate photoreceptors at the surface level, green light penetrates deeper into plant tissues, thereby affecting internal photoreceptors [27, 28].

Guo, Zhou [29] demonstrated that green LED lighting at 25 μmol·m$^2$·s$^{-1}$ for 12 d inhibited malondialdehyde accumulation and maintained higher chlorophyll content in cabbage (*Brassica oleracea* var. capitata) by regulating glucosinolate metabolism, while Jin, Yao [30] found that broccoli florets (*Brassica oleracea*. ssp. *italica*) exposed to green LED light had higher sulforaphane content compared to those stored in darkness or fluorescent lighting for 2 d at 25°C. Castillejo, Martínez-Zamora [27] found that green LED increased total glucosinolate content by 34.5% although green LEDs had no impact on the total phenolic content on ready-to-eat broccoli sprouts (*Brassica oleracea* var. italica) during 15 days at 5°C compared to traditional storage in darkness. These findings suggest that green LED light may influence biosynthetic pathways through pigment photoreceptor proteins, such as cryptochromes and phytochromes, leading to the accumulation of bioactive compounds, which are crucial for enhancing the nutritional quality of postharvest vegetables [27, 28].

Jin, Yao [30] demonstrated that green LED light (12 μmol m$^{-2}$ s$^{-1}$) increased total phenolic content and DPPH radical scavenging activity in broccoli florets. This effect was linked to enhanced activity of phenylalanine ammonia-lyase (PAL), a key enzyme in the phenylpropanoid pathway responsible for the synthesis of phenolic compounds. Green LED light has been shown to increase glucosinolate levels in broccoli, indicating that light quality and dosage can significantly influence phytochemical synthesis and overall nutritional quality [30].

Pennisi, Orsini [31] found that green LED light helped maintain color and pigment content in rocket leaves (*Diplotaxis tenuifolia*) during storage, suggesting that light quality, particularly green light, is a critical exogenous factor in regulating senescence. The capacity of harvested leaves to respond to light stimuli, mediated by photoreceptors, underscores the importance of proper light management to balance functional metabolism, such as maintaining chlorophylls and carotenoids, with the natural progression of leaf senescence [31, 32].

Green wavelengths, specifically 500–560 nm, can induce stomatal closure more effectively than red and blue light, significantly reducing transpiration and moisture loss in spinach, kale, basil, and lettuce plants, thereby maintaining better visual quality [33, 34]. Despite these promising results, there is still limited understanding of which specific green spectra are most

effective in preserving or enhancing the health-promoting properties of postharvest vegetables, especially anthocyanin and flavonoid. The aim of this work was to build on this foundational research by assessing the impact of two monochromatic green LED lights (500 nm and 530 nm) in comparison to white LED lighting (400–700 nm) and dark storage on the nutritional quality of lettuce heads stored at 5˚C for 14 d. Antioxidant activity, anthocyanins, flavonoids, chlorophylls, carotenoids, and total soluble solids were measured to better understand how these wavelengths influenced the nutritional quality of lettuce during postharvest storage. By improving postharvest storage conditions using green LEDs, it may be possible to reduce waste and provide consumers with higher quality produce.

## Results

### Effects of postharvest light spectra on antioxidant system of lettuce

The antioxidant activity of the lettuce leaf samples decreased following 14 d of cold storage compared to the initial evaluation ($p < 0.01$). Remarkably, the reduction in antioxidant activity was less noticeable in lettuce exposed to green LEDs compared to those stored in the dark (Fig 1A). Samples stored in the dark experienced a drastic decrease in antioxidant activity, declining by 94.5% compared to their initial values. In contrast, light-exposed samples

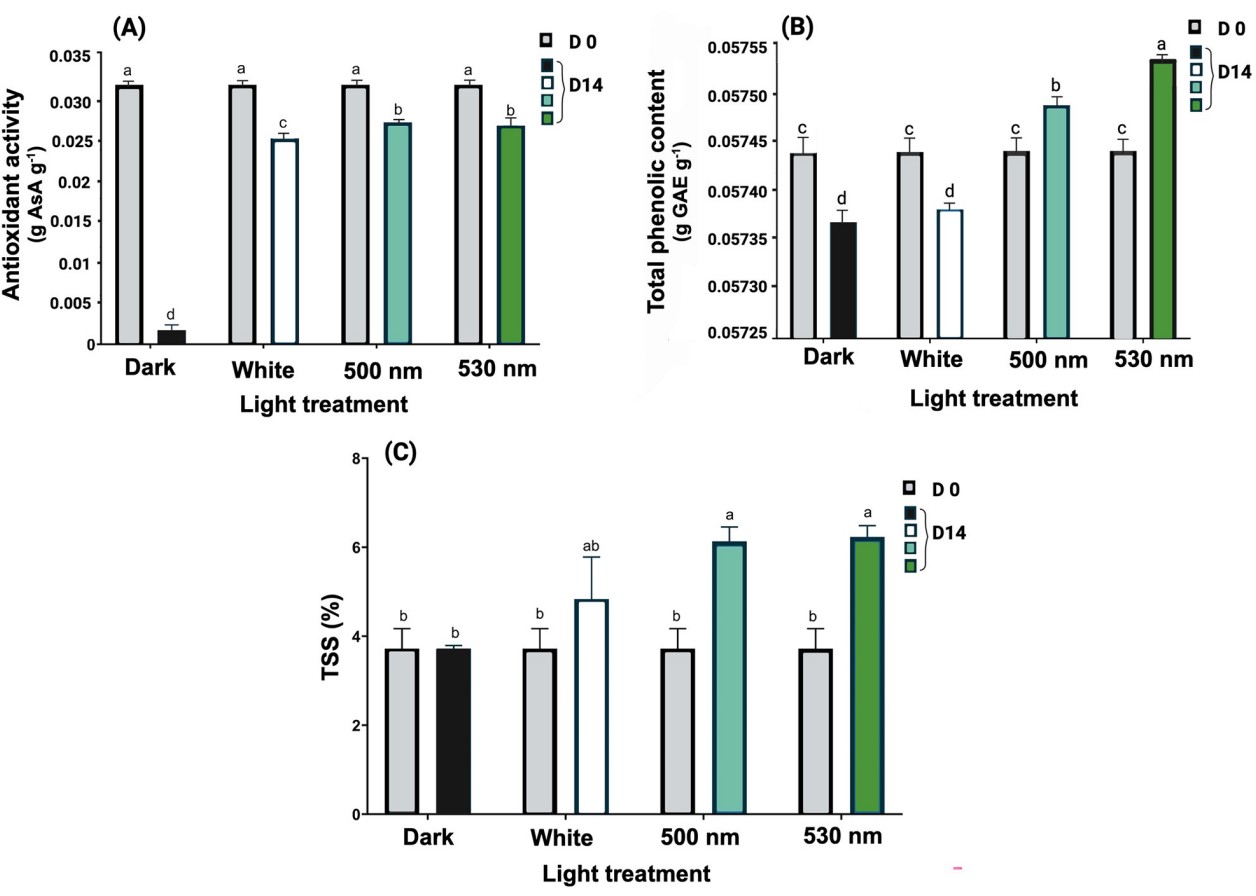

**Fig 1. Effect of postharvest light spectra on antioxidant activity (A), total phenolic content (B) and Total soluble solid [TSS; (C)] in lettuce leaf samples stored under different light conditions, including 530 nm and 500 nm green LEDs, white LEDs (400–700 nm), and dark storage for a duration of 14 d at 5˚C.** Values represent the means of three replications ± SE. Data in columns with different letters are significantly different $p < 0.01$.

maintained at least 80% of their antioxidant activity. Specifically, white LED-treated samples showed a reduction of 20.6%, while samples exposed to 530 nm and 500 nm green LEDs exhibited a 15.7% and 14.4% decrease, respectively. Samples exposed to 500 nm, 530 nm, and white light retained respectively 15.7, 15.5, and 14.6 times higher antioxidant activity compared to the antioxidant activity of lettuces stored in the dark at the end of the experiment. Overall, green light exposure prevented a drastic decrease in total antioxidant activity in lettuce leaves during postharvest cold storage.

The total phenolic content of lettuce stored in the dark and under white LEDs decreased to similar levels after 14-d storage, compared to their initial values (0.057 g/g dry matter- $p < 0.01$). In contrast, total phenolic content increased significantly in lettuce stored under 530 nm and 500 nm green LEDs compared to their initial values ($p < 0.01$). Furthermore, green LED-stored lettuce exhibited a significant increase for 530 nm and 500 nm in total phenolic content compared to its content in dark-stored samples ($p < 0.01$). However, there was no significant difference between white LED- treated samples and those kept in the dark (Fig 1B).

The impact of the experimental LED spectra on the total soluble solid (TSS) content in lettuce is illustrated in Fig 1C. No significant differences were observed between the dark and white treatments and their initial values. However, among the different light treatments, the TSS of lettuce exposed to 530 nm green LEDs demonstrated the highest increase (67.5% increase), followed by 500 nm green LEDs (64.8% increase) compared to their initial value and dark storage. These results indicate that green light notably increased the TSS content of lettuce in comparison with the dark storage conditions and their initial values ($p < 0.05$).

## Effects of postharvest light spectra on protective pigments of lettuce

Total anthocyanin content in all lettuce samples significantly decreased during postharvest storage ($p < 0.01$). The 530 nm green LEDs prevented a considerable decrease in anthocyanin content, exhibiting only a 36.4% decline compared to its initial value (0.53 mg/g dry matter). Anthocyanin content in lettuce stored with 530 nm green LEDs notably contained a higher anthocyanin content ($p < 0.01$) in comparison to samples exposed to the dark (2.28 times higher). There was no significant difference between the lettuce samples exposed to 500 nm green LEDs and dark storage, showed 72.1% and 75.2% decrease compared to their initial values, respectively. However, the samples stored under white light experienced the most drastic decline, with a 91.4% decrease, in comparison with the initial values and a 69.2% decline compared to the samples stored in the dark (Fig 2A).

Total carotenoid content of lettuce stored under both the 530 nm and 500 nm green LEDs increased significantly to similar levels compared to their initial values ($p < 0.01$) after 14 d of storage (Fig 2B). This increase was 12.7% for 530 nm and 11% for 500 nm in comparison with their initial value, and 26.2% and 24.2% in comparison with the dark-stored samples. Conversely, lettuce samples stored with the white LEDs or dark experienced a significant reduction of 8.2% and 10.7% in total carotenoid content compared to the initial value ($p < 0.01$), respectively. There was no significant difference between the carotenoid content of dark and white light-exposed samples.

Lettuces stored under both green LED spectra exhibited a significantly higher total flavonoid content compared to the white light treatment and dark storage. White LEDs had no significant effect on the total flavonoid content of lettuce leaf during 14 d of postharvest storage. Total flavonoid content at d 14 of postharvest storage was increased by 2.6 and 2.2 folds as a result of lettuce leaf exposure to 530 nm and 500 nm green lights, respectively (Fig 2C). The samples stored in the dark had a 1.3-fold increase in the total flavonoid content. Furthermore,

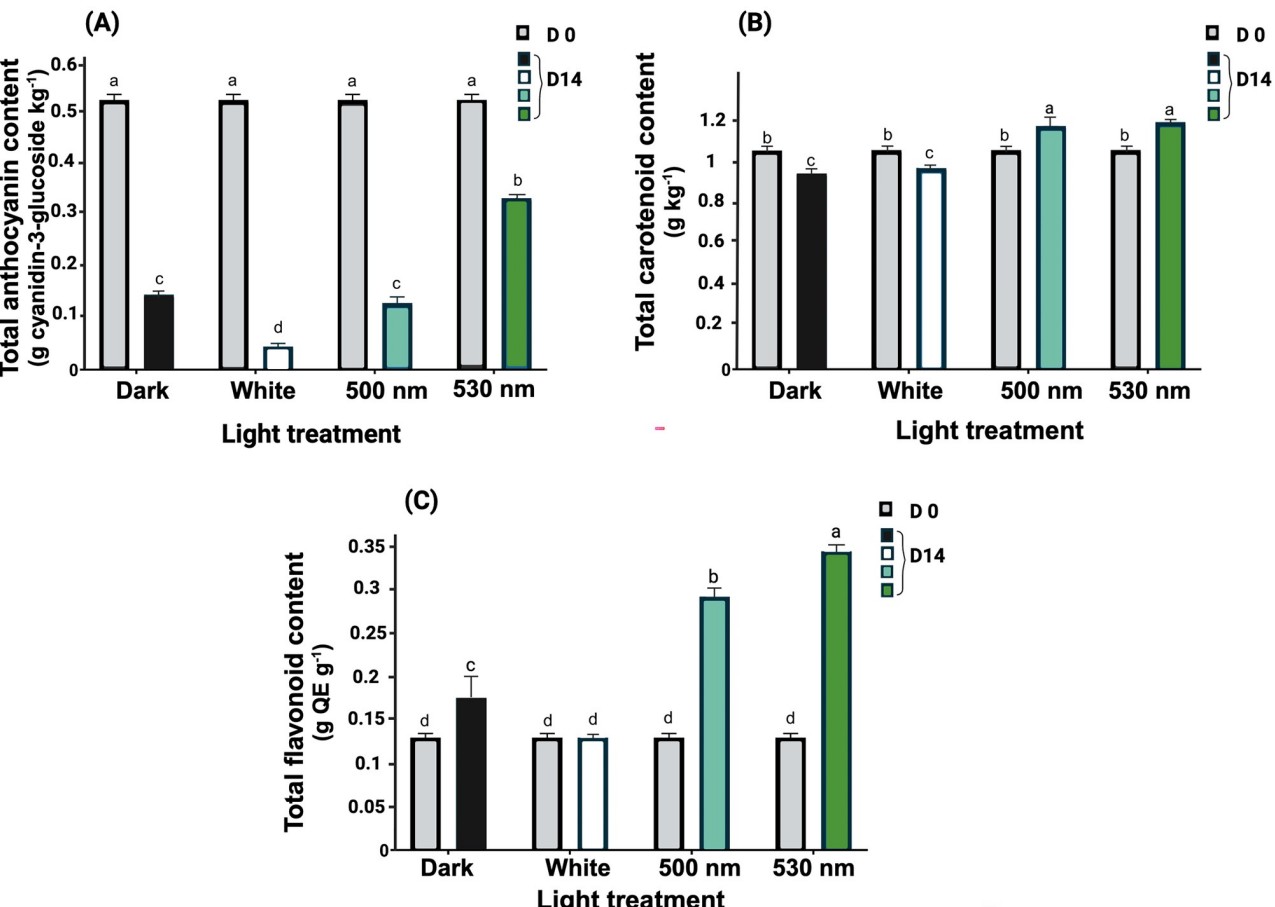

**Fig 2. Effect of postharvest light spectrum on protective pigments including anthocyanins (A), carotenoids (B), and total flavonoids (C) in lettuce leaf samples stored under different light conditions, including 530 nm and 500 nm green LEDs, white LEDs (400–700 nm), and dark storage for a duration of 14 d at 5°C.** Values are the means of three replications ± SE. Data in columns with different letters are significantly different p < 0.01.

530 nm and 500 nm exposure caused respectively 95% and 65% increase in flavonoid content compared to its content in the samples stored in the dark, whereas the white LED treatment caused 25.96% decrease compared to the flavonoid content of dark-stored samples following 14 d of postharvest storage.

## Effects of postharvest light spectra on photosynthetic pigments in lettuce

Chlorophyll *a* content decreased significantly in both dark and white light-exposed samples during storage (p < 0.01). Interestingly, lettuce stored with green 500 nm LEDs exhibited a notably higher chlorophyll a content compared to the initial value (11.3%). Compared to the initial value, the chlorophyll *a* content in samples stored in the dark and under white LEDs decreased by 12% and 9.3%, respectively. Conversely, there was no changes between those stored under the 530 nm green LEDs and their initial value. Lettuce leaf samples exposed to 500 nm and 530 nm green LEDs displayed respectively 26.4% and 16.0% higher chlorophyll *a* content compared to samples stored in dark, while the white LED treatment showed no significant difference with chlorophyll *a* content in dark-exposed samples (Fig 3A).

All lettuce samples displayed statistically similar chlorophyll *b* content, except for samples exposed to 530 nm green light, where the chlorophyll *b* content decreased by 29% compared to

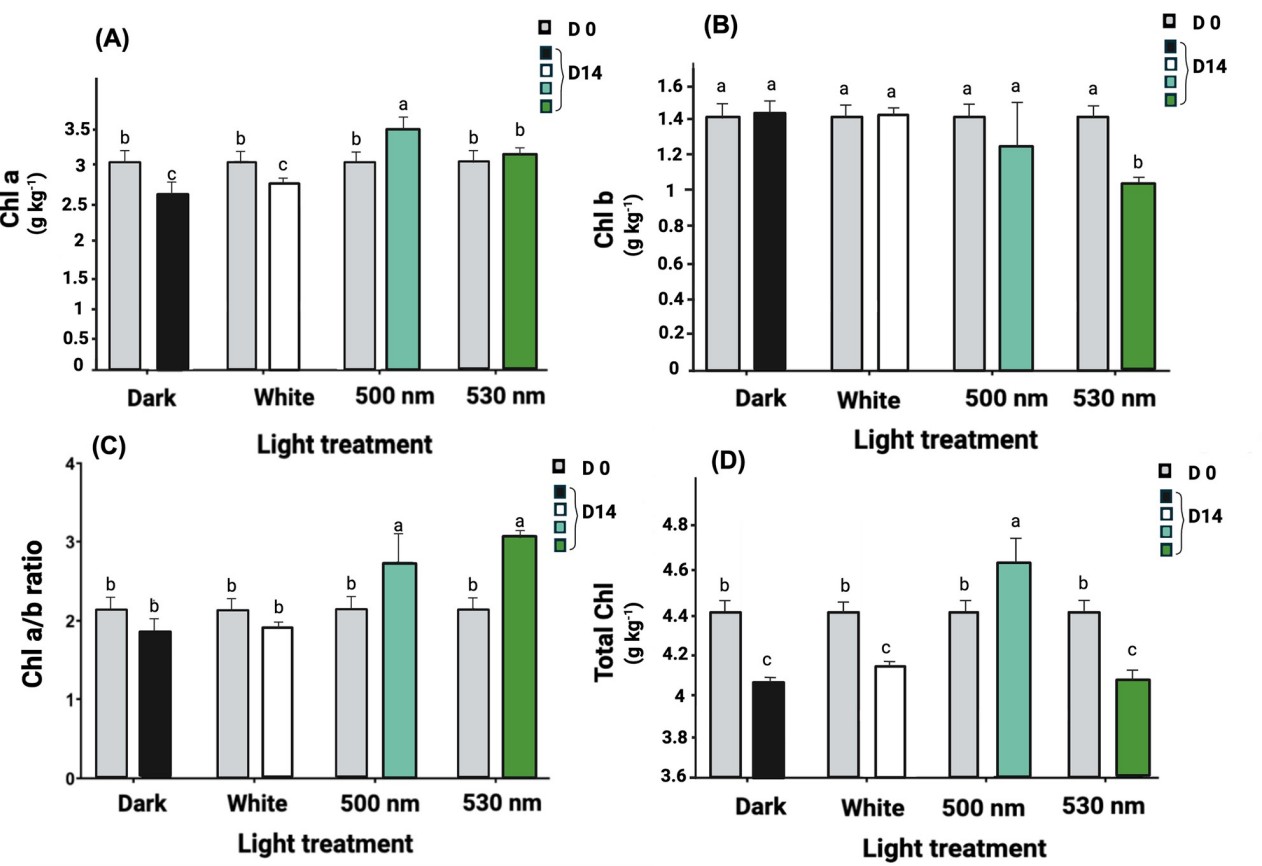

**Fig 3. Effect of postharvest light spectrum on photosynthetic pigments including chlorophyll (Chl) *a* (A), Chl *b* (B), Chl *a/b* ratio (C), and total Chl content in lettuce leaf samples stored under different light conditions, including 530 nm and 500 nm green LEDs, white LEDs (400–700 nm), and dark storage for a duration of 14 d at 5˚C.** Values are the means of three replications ± SE. Data in columns with different letters are significantly different (p < 0.01).

both their initial values or for the dark storage (p < 0.05). In the case of the 500 nm green LEDs, there was a 10% reduction in the chlorophyll *b* content compared to their initial value for the dark storage; however, this reduction was not significantly different (Fig 3B).

The Chl *a/b* ratio in lettuce leaf samples treated with dark or white LEDs exhibited 13.2% and 10.0% decreases, respectively, in comparison with their initial ratio (Fig 3C). However, these reductions were not statistically significant compared to the initial values. Conversely, lettuce treated with 530 nm and 500 nm green LEDs showed 43.6% and 27.1% increases, respectively, in Chl *a/b* ratio in comparison to their initial ratios (p < 0.01). Compared to the dark condition, the 530 nm and 500 nm LEDs had a 65.6% and 46.6% increase in Chl *a/b* ratio (p < 0.01), respectively, yet there was no significant difference between these two green LEDs. In contrast, the white LED and dark treatments showed no significant differences with each other or with their initial ratios.

Compared to the initial content, lettuce samples exhibited a notable reduction in total chlorophyll content (7.7%, 7.6%, and 6.1% for 530 nm, dark storage, and white LED treatments, respectively), with the exception of those stored under 500 nm green light (Fig 3D). In the case of the 500 nm green LEDs, a significant increase (4.8%) in total chlorophyll content was detected compared to the initial content (p < 0.01). There was no significant difference

between the total chlorophyll content in samples treated with white and 530 nm green LEDs and those stored in the dark. In contrast, the 500 nm green LEDs led to a 13.5% higher total chlorophyll content compared to samples stored in dark (p < 0.01).

### Principle component and correlation matrix analyses for the bioactive compounds and color of lettuce as affected by postharvest lighting environment

A matrix correlation analysis was conducted to examine the relationship between bioactive compounds and color attributes in the lettuce leaf samples stored under different light conditions, including green LEDs with peaks at 500 nm and 530 nm, white LEDs, and dark storage. The Pearson correlation coefficients exhibited intricate linkages among the variables under investigation (Fig 4).

Principal component analysis (PCA) revealed information regarding the levels of bioactive chemicals and color characteristics in lettuce following 14-d storage period at 5°C, while being subjected to different light treatments (Fig 5). The PCA biplot exhibits a notable clustering pattern, indicating a clear differentiation among treatments. Notably, the treatments involving green LEDs with peak wavelengths at 500 nm and 530 nm exhibit a prominent separation. The graph provides a visual representation of the key components that contribute to the observed changes. The results of the subsequent clustering analysis corroborate and extend these findings by explicitly delineating sample groupings based on similarity. Under dark storage conditions, both at initiation (day 0) and end of the storage period (day 14), samples formed distinct clusters (cluster 1 and cluster 2, respectively), indicating a shift in biochemical composition over time. Similarly, samples exposed to white LEDs exhibited cluster transitions from day 0 to day 14, further elucidating the evolving nature of the lettuce characteristics. Of particular interest are the samples treated with green LEDs at 500 nm and 530 nm. At day 0, leaf samples stored under both 500 nm and 530 nm green LEDs aligned with Cluster 1, signifying a certain degree of similarity with the dark and white conditions. However, by day 14, a notable divergence occurred, with 500 nm green LEDs at day 14 and 530 nm green LEDs at day 14 forming a distinct cluster 3. This divergence, highlighted by increased distances to centroids, underscores the profound impact of these specific light treatments on the biochemical profiles of the lettuce during the storage period.

## Discussion

This study aimed to explore the impact of green LEDs, white LEDs and dark storage on the maintenance of nutritional value in lettuce. Levels of chlorophylls, carotenoids, flavonoids, antioxidants, anthocyanin, phenolic compounds, soluble solids, and color characteristics were compared in lettuce both before and after the 14-d storage period.

### Antioxidant capacity

LED lighting has proven a valuable tool for preserving phenolic compounds and antioxidant activity in harvested produce [31, 35]. Our research indicates that employing green LEDs (specifically with a peak at 530 nm) effectively increased total phenolic content, thereby leading to a significant retention of total antioxidant activity in lettuces, as compared to dark storage and other light treatments. Similar outcomes have been reported by Lee, Ha [24], where cabbage stored with green LED irradiation led to an increase in the total polyphenol content and total antioxidant activity in comparison to dark storage, aligning with findings reported by Jin, Yao [30] for broccoli and Samuolienė, Brazaitytė [36], Samuolienė, Sirtautas [37] for baby leaf

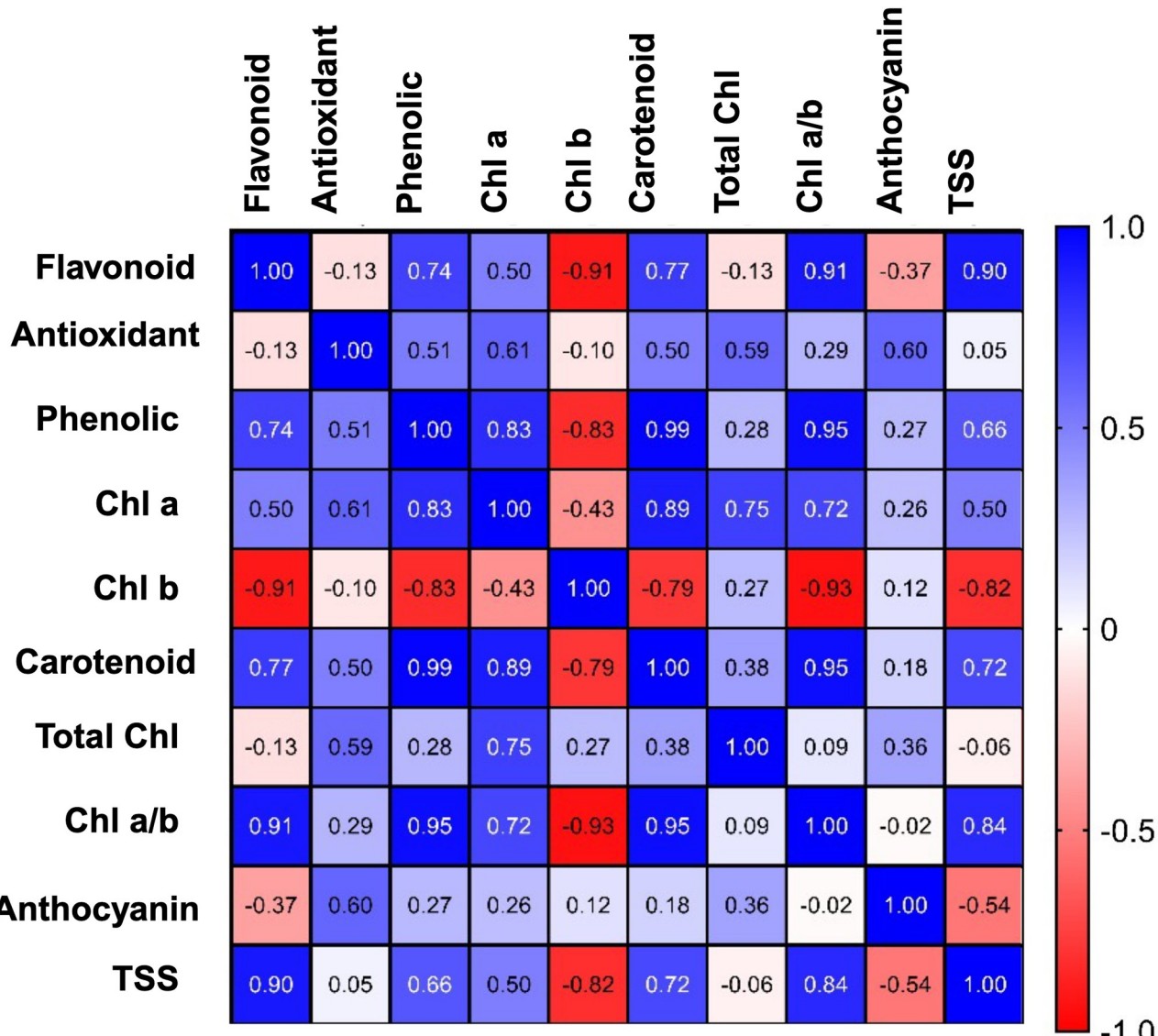

**Fig 4. The correlation matrix for the effect of postharvest light spectrum on bioactive compounds of lettuces.** The samples were exposed to 530 nm and 500 nm green light, white light (400–700 nm), and dark for a duration of 14 d at 5°C.

lettuce. This phenomenon is likely attributed to the induction of phenylalanine ammonia-lyase activity, a crucial enzyme in the phenylpropanoid pathway, in lettuce exposed to green light [38]. Higher retention of antioxidant activity in green LED-treated lettuce at 530 nm in our work could be mainly ascribed to their higher levels of total phenolics, flavonoids or anthocyanins in comparison to dark storage, which is in accordance with Samuolienė, Brazai-tytė [36]. It has been suggested that combining different antioxidants can create synergistic effects, potentially enhancing their overall antioxidant capacity [39]. Further investigation and experimentation, including molecular and physiological studies, could provide a more detailed understanding of the mechanisms underlying these exceptions.

Kasim and Kasim [1] observed that green light significantly contributed to the preservation of chlorophyll content and TSS in lettuce when compared to a dark environment after a 21-d

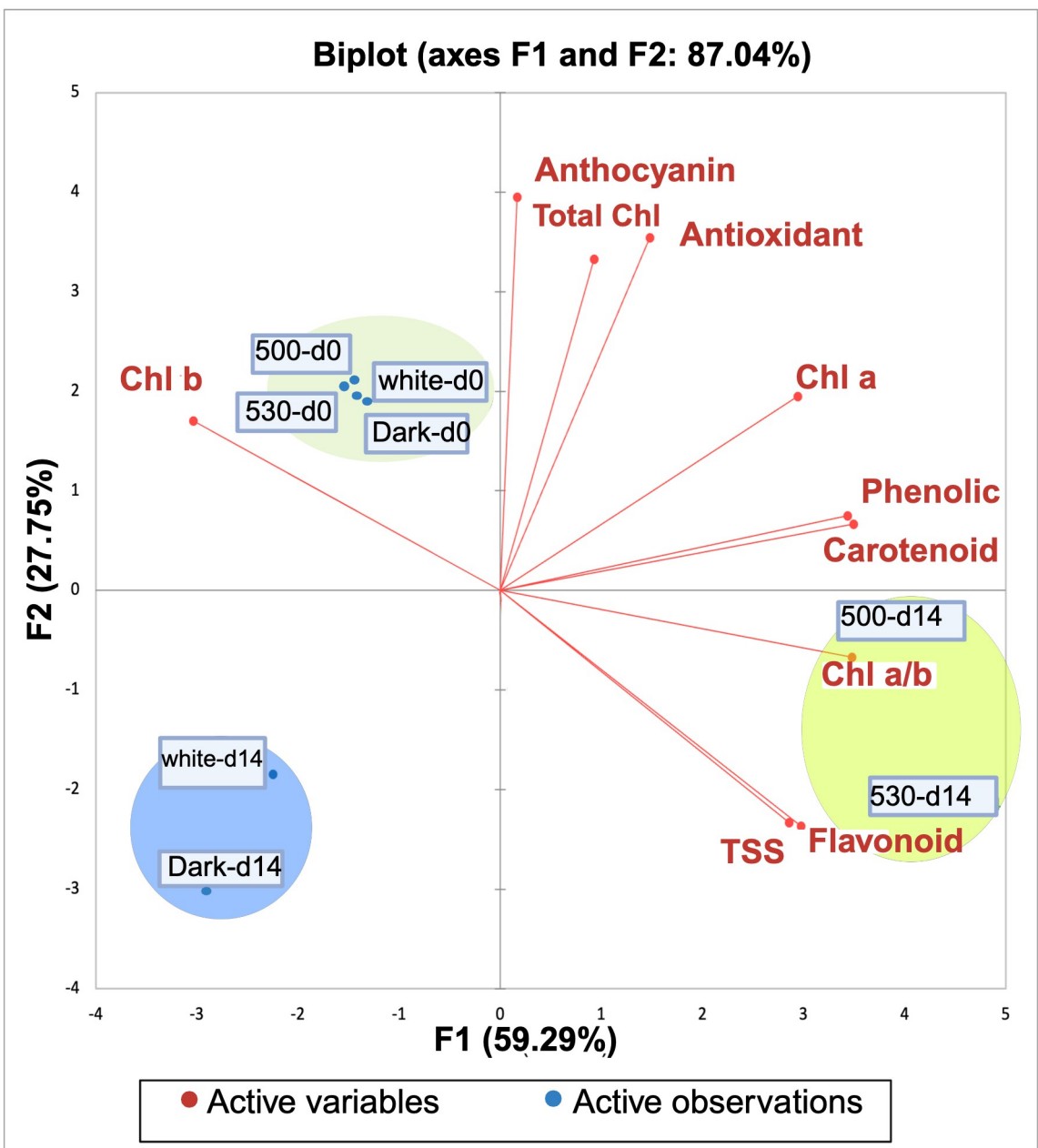

**Fig 5. Principle component analysis for the effect of postharvest light spectrum on bioactive compounds of lettuces.** The samples were exposed to 530 nm and 500 nm green light, white light (400–700 nm), and dark for a duration of 14 d at 5˚C.

storage. Kim, Lee [40] indicated a gradual increase in TSS in light-treated groups, surpassing the levels observed in the dark for strawberries after 4-d storage. According to Zhan, Hu [41] illuminating fresh broccoli with light during the initial storage phase conserved or stimulated the production of photosynthetic products. The outcomes of the current study regarding lettuce's TSS align with these previous findings.

## Protective pigments

Data presented herein show that 530 nm green LEDs significantly mitigated the loss of anthocyanin content compared to dark storage, emphasizing its potential significance in preserving these vital compounds. This effect may be attributed to anthocyanins' absorption efficiency in the blue to green spectrum, particularly around 500–550 nm [29]. Gitelson, Merzlyak and Chivkunova [42] observed that the absorbance of maple (*Acer platanoides*) and cotoneaster (*Cotoneaster alaunica*) leaves at approximately 550 nm and within the 400–600 nm band is directly proportional to anthocyanin content. Higher anthocyanin levels result in increased absorbance within this green spectral range [37]. Also, 530 nm wavelength might closely align with the optimal absorption range of anthocyanins, potentially activating protective mechanisms or biosynthetic pathways that stabilize these compounds during storage. This specific wavelength might enhance the activity of key enzymes involved in the anthocyanin biosynthetic pathway, such as chalcone synthase (CHS) or dihydroflavonol 4-reductase (DFR), leading to higher retention of anthocyanins [42, 43]. Although the effect of green light at 500 nm was relatively modest, showing similar anthocyanin content to dark conditions, it still had a positive influence when compared to white light. Red light exposure during growth significantly boosts the anthocyanin content in lettuce compared to white light treatment [44], and green and blue LEDs tend to enhance anthocyanin content in herbs [39, 45]. The implications of these findings underscore the intricate relationship between light quality and preservation of essential compounds such as plant anthocyanins, highlighting potential pathways for optimizing their accumulation through various metabolic responses [43, 46].

## Postharvest green light exposure increased total carotenoid content in lettuce leaves

Research indicates that chlorophyll and carotenoid content tends to diminish during storage [47]. Light quality plays a crucial role in determining carotenoid accumulation [48]. Exposure to red light leads to elevated lycopene content in tomatoes and for broccoli, exposure to blue light at 20 μmol m$^{-2}$ s$^{-1}$ at 22°C for 3 d resulted in increased carotenoid levels [49]. Lamb lettuce exposed to warm white light exhibited a slower decline in carotenoid content [50]. Loi, Liuzzi [14] showed no significant difference between green (524 nm), blue (467 nm), red (625 nm) and yellow (587 nm) at an intensity of 24 μmol m$^{-2}$ s$^{-1}$ and dark control after 20 d of storage of broccoli. In this work, the green LED-treated samples exhibited a greater quantity of total carotenoids compared to the dark control samples (26% and 24% for 530 and 500 nm, respectively), their initial value or white LED-treated lettuce. This increase in pigment content may be attributed to the positive influence of green light on the synthesis of carotenoids and chlorophylls [51]. Consistent with our results, Pennisi, Orsini [31] was found that the carotenoid content remained higher (21% higher in comparison with dark-stored samples) after 10 d of storage in red chards stored under 517 nm green light with an intensity of 35 μmol m$^{-2}$ s$^{-1}$. A low light intensity (10 μmol m$^{-2}$ s$^{-1}$) was used in this work, which could explain the higher pigment content after 14 d of exposure to green LEDs light. Hasperué, Rodoni [51] observed a positive effect with white-blue LEDs on postharvest carotenoid accumulation in Brussel sprouts (*Brassica oleracea* var gemmifera).

## Total flavonoid content enhanced by green light during postharvest storage

The total flavonoid content in lettuce stored under green LEDs was significantly higher compared to lettuce stored in the dark or under white light. The increased content of flavonoids in the green-treated samples may be attributable to the light quality. Previous studies have shown

similar results in which the flavonoid synthesis stimulated by exposure to light as a protective response [51–53]. Alterations in reactive oxygen species (ROS) stimulate the biosynthesis of flavonoids, which have the capability to scavenge singlet oxygen and stabilize chloroplast membranes [51, 54]. Kanazawa, Hashimoto [52] tested UV-A, UV-B lamps and green LEDs on parsley (*Petroselinum crispum*), Indian spinach (*Basella rubra*) and garden pea sprout (*Pisum stivum*) storage in a home refrigerator. They demonstrated that light with wavelengths near the absorption λmax of phenylpropanoids, particularly around 503 nm for pelargonidin glycoside, can induce flavonoid synthesis. Additionally, the expression of genes involved in phenylpropanoid biosynthesis was found to induce flavonoid synthesis [52, 55]. For vegetables with anthocyanin content and a λmax around 503 nm, green light in the 500−600 nm range is effective in inducing flavonoid synthesis.

## Photosynthetic pigments

Senescence in harvested green organs typically involves the extensive breakdown of chlorophylls and chloroplast proteins [18, 31, 56]. These responses are influenced by light quality and intensity [32]. Excessive light intensity can lead to photo-oxidative damage, resulting in the degradation of chlorophylls and carotenoids [31, 48]. In this study a low level light (10 μmol $m^{-2}$ $s^{-1}$) was used to prevent photo-oxidative damages to the leaves, as described previously [57]. Under the experimental conditions employed, total chlorophyll content of all lettuce samples exhibited a decrease, with exception of those subjected to storage under green (500 nm) LEDs, which displayed a notable augmentation in total chlorophyll content. The observed increase in chlorophyll content and intensified green coloration in the lettuce following 14-d cold storage is noteworthy and consistent with consumer preferences for high-quality produce. The increased concentration of pigments can be ascribed to the beneficial impact of green light, particularly at a wavelength of 500 nm, on the process of chlorophyll synthesis and the promotion of chlorophyll metabolism [58]. This effect is influenced by the reduced electron transport and/or the heightened abundance of divinyl chlorophyllide *a*, a significant precursor in the biosynthesis of chlorophyll [59]. The potential influence of low-level light on chlorophyll-related genes and enzymes during post-harvest handling has been investigated by several researchers [59–61]. According to Zhou, Zuo [60], the upregulation of the BrHEMA1 gene, which is involved in chlorophyll synthesis, is observed in samples treated with LED light. This upregulation leads to an increase in chlorophyll production and a decrease in the expression of genes responsible for chlorophyll breakdown. Nevertheless, it is plausible that there exist alternative explanations for the observed augmentation in chlorophyll levels in lettuce exposed to green (500 nm) LED lighting following a 14-d storage period.

## Postharvest green light (500 nm) exposure enhanced the Chl a content in lettuce

The observed rise in Chl *a* concentration following postharvest exposure to 500 nm light may be attributed to the lettuce leaves' adaptive responses aimed at maximizing photosynthetic efficiency. This reaction may encompass the metabolic processes that promote the synthesis of Chl *a* and the genetic variety present in the plant [62]. The observed reduction in Chl *b* concentration subsequent to exposure to 530 nm light may be attributed to the preferential absorption characteristics of Chl *b*. Chl *b* predominantly absorbs light within the blue and red-orange regions of the electromagnetic spectrum, while exhibiting lower efficiency in absorbing green light (about 530 nm), and absorption of Chl *b* may be negatively affected by exposure to the 530 nm wavelengths. Another potential factor contributing to this phenomenon could be modifications in chlorophyll *a* to chlorophyll *b* ratio, which serves to enhance the efficiency of

photosynthetic processes. It is worth noting that a significant increase in the Chl a/b ratio of lettuce was observed after storage under green LEDs, particularly with a wavelength of 530 nm, compared to lettuce stored under white LEDs or in the dark. This implies that the utilization of green LED light could potentially improve the postharvest storage and preservation of an ideal Chl *a/b* ratio, boosting light absorption and the efficiency of photosynthesis. In the context of postharvest settings, it is often preferable for the Chl *a/b* ratio to have a larger value, since this signifies a prevalence of Chl *a*. This particular form of chlorophyll has been observed to give more favorable outcomes. Furthermore, previous studies have provided evidence that plants have the ability to modify their chlorophyll concentrations in response to variations in light intensity and quality [50]. This adaptation results in an elevated ratio of Chl *a/b* ratio to optimize the capture of red light [63], and in this way affecting both leaf light absorption and visual quality. Alternatively, in situations with limited light availability, the Chl *a/b* ratio may be decreased to improve the absorption of blue light [64, 65]. According to Lee, Ha [24], the distribution of chlorophyll inside plants leads to variations in the absorption of light wavelengths. Additional empirical research is required to validate these mechanisms and delve into their implications within the context of lettuce preservation. Previous research indicates that the use of green light has a beneficial effect on the chlorophyll content of postharvest vegetables. The application of green LED light either preserves or boosts chlorophyll content in broccoli heads (*Brassica oleracea* ssp. *italica*) [14, 30]. Continuous postharvest illumination with green light results in elevated overall chlorophyll content in cabbage (*Brassica oleracea* var. *dongdori*) [24], as well as preserving of chlorophyll content in rocket (*Diplotaxis tenuifolia*) [31].

## Relationship between preserving different bioactive compounds in lettuce

PCA and correlation matrix analyses performed in this work offer new insight into the complex biochemical dynamics of lettuce subjected to different light spectra. PCA revealed distinct clustering patterns that not only differentiate the biochemical states of lettuce subjected to various light treatments but also highlight the profound impact of specific green wavelengths (500 nm and 530 nm) on metabolic processes. The clear separation of clusters corresponding to these green LED treatments from those associated with dark and white LED treatments suggests a fundamentally different metabolic trajectory. This divergence is likely driven by the selective activation of photoreceptors that are responsive to green light, which in turn modulates key biochemical pathways involved in pigment preservation and secondary metabolite synthesis [52, 64, 66].

The clustering of samples under green LED treatments can be interpreted as a manifestation of enhanced metabolic stability. Green light, particularly at 530 nm, appears to effectively modulate the chlorophyll degradation pathway, which is typically accelerated during postharvest senescence. By stabilizing chlorophyll a, which is more prone to degradation compared to chlorophyll b, the 530 nm green light helps maintain photosynthetic capacity, even under storage conditions [47, 59, 61]. This retention of chlorophyll a not only preserves the photosynthetic machinery; it additionally promotes the continued synthesis of secondary metabolites, such as flavonoids and phenolics, which are crucial for mitigating oxidative stress [67, 68]. The strong positive correlations between the chlorophyll a/b ratio and the levels of flavonoids and phenolics, as revealed by the correlation matrix, suggest a coordinated regulation of these compounds under green light. The chlorophyll a/b ratio is an indicator of the balance between light-harvesting complexes and the core photosynthetic reaction centers. A higher ratio, particularly under green LED treatments, indicates a preferential retention or synthesis of chlorophyll a, which is more efficient in capturing and converting light energy [1, 66, 69]. This

enhanced photosynthetic efficiency likely provides the energy required for the biosynthesis of flavonoids and phenolics, which serve dual roles as antioxidants and photoprotective agents.

The specific wavelengths of green light (500 nm and 530 nm) appear to activate distinct sets of photoreceptors, such as cryptochromes that, are known to regulate the expression of genes involved in flavonoid biosynthesis. The upregulation of enzymes like chalcone synthase (CHS) and dihydroflavonol 4-reductase (DFR) under green light can lead to increased flavonoid accumulation, as observed in the positive correlations with chlorophyll a/b ratios [52, 53]. This synergistic effect between pigment retention and secondary metabolite production highlights the potential of green light to enhance the nutritional quality of lettuce during storage [29].

The negative correlation between chlorophyll a and chlorophyll b suggests a finely tuned regulatory mechanism that balances the degradation and synthesis of these pigments in response to green light. The selective preservation of chlorophyll a under 530 nm green LED treatment might be linked to its role in maintaining the integrity of the photosystem II reaction center, which is critical for sustaining photosynthetic activity during stress conditions, such as prolonged storage. The differential response of chlorophyll b, which is more closely associated with light-harvesting complexes, may reflect a shift in the allocation of resources towards maintaining core photosynthetic functions [47].

The PCA analysis revealed distinct clustering patterns, particularly for the 530 nm green LED treatment, indicating a unique biochemical state induced by this wavelength. This suggests that 530 nm green light not only slows the degradation of key pigments and metabolites but also promotes a metabolic state enriched in bioactive compounds. This may result from the activation of specific stress-responsive pathways, representing an adaptive response that optimizes the lettuce's biochemical composition for extended storage. These findings underscore the potential benefits of utilizing green 500 nm and 530 nm LED light as a tool to maintain and improve the nutritional composition of stored lettuce, driving advancement of postharvest storage systems, with the potential to bring about changes in the practices within the agricultural and food industries. These data support previous findings, which showed the effectiveness of green wavelengths in keeping stomata in a closed state in both basil (*Ocimum basilicum*) and lettuce, suggesting its potential in maintaining postharvest quality and prolonging the shelf life of leafy vegetables [34].

## Conclusion and future perspectives

Exposure of lettuce to light throughout its storage period can yield advantageous outcomes for a duration of 14 d at 5°C. Noteworthy findings include a significant increase in antioxidant activity, where samples exposed to 500 nm and 530 nm green LEDs displayed 1474.5% and 1451.8% (approximately 15.7 and 15.5 times) higher activity, respectively, compared to those stored in the dark. Substantial enhancements in total phenolic content were additionally observed, and green 530 nm and 500 nm treatments resulted in 67.5% and 64.8% increases in total soluble solids compared to dark storage. Exposure to 530 nm green light further led to a 2.28-fold higher anthocyanin content, a 26.2% increase in carotenoids, and a 95% rise in flavonoid content compared to dark-stored samples. Chlorophyll-related parameters showed increases of 26.4% and 16.00% in chlorophyll a content for 500 nm and 530 nm, respectively, along with a 65.6% and 46.6% rise in the Chl a/b ratio. The 500 nm treatment resulted in a 13.5% higher total chlorophyll content compared to dark-stored samples. Further investigation into how storage under green LED in the cold storage affects quality parameters such as moisture loss, sensory analysis, respiration rate, enzymology and microbiology in leafy greens is warranted, and additional work will explore different combinations of green light with modified atmosphere packaging.

## Material and methods

### Plant material

Romaine lettuce (*Lactuca sativa* cv. Breen) was harvested from La Boîte Maraîchère (LBM) a local commercial vertical hydroponic farm in Laval, Quebec, Canada. The lettuce was promptly transported to the laboratory within 1 h of harvesting. Fresh lettuce heads, exhibiting uniform size and color, and showing no signs of disease or mechanical damage, were selected for the experiment.

### LED lighting

Two different green LED assemblies with peak wavelengths at 500 nm and 530 nm, along with white LEDs (UTechnology, Calgary, Alberta, Canada) were used for the experiment (Fig 6). The photosynthetic photon flux density (PPFD) necessary for post-harvest preservation is typically minimal, spanning from 10 to 80 $\mu mol\ m^{-2}\ s^{-1}$ as indicated in multiple research studies [69]. The light intensity employed in this experiment was set at 10 $\mu mol\ m^{-2}\ s^{-1}$ PPFD, based on our preliminary research [34]. This intensity level was measured with a quantum meter

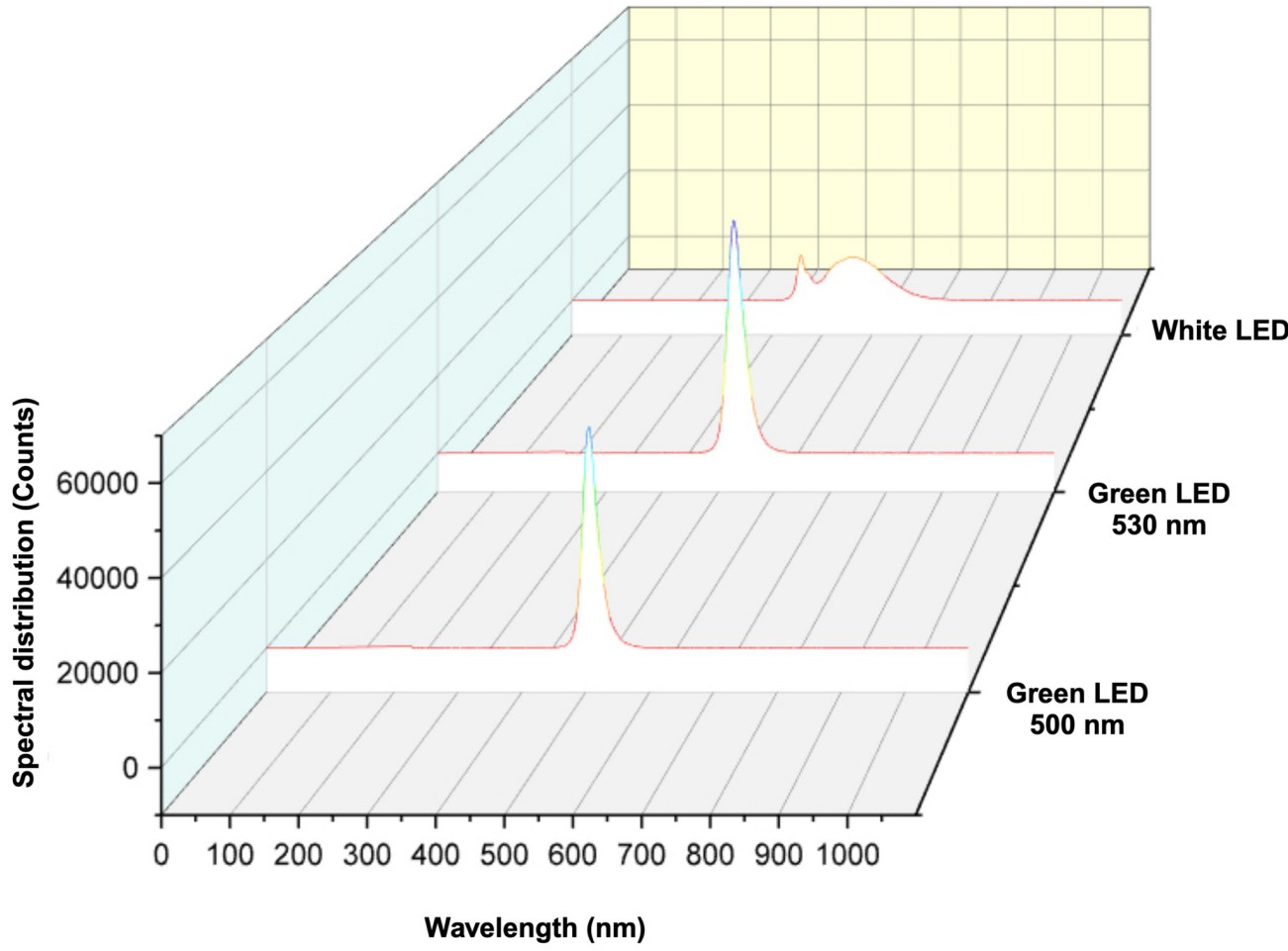

**Fig 6. Spectral distribution and peaks of three LED assemblies used in this study including green LEDs with peaks at 500 nm and 530 nm, as well as white LEDs (400–700 nm).** LED: light emitting diode.

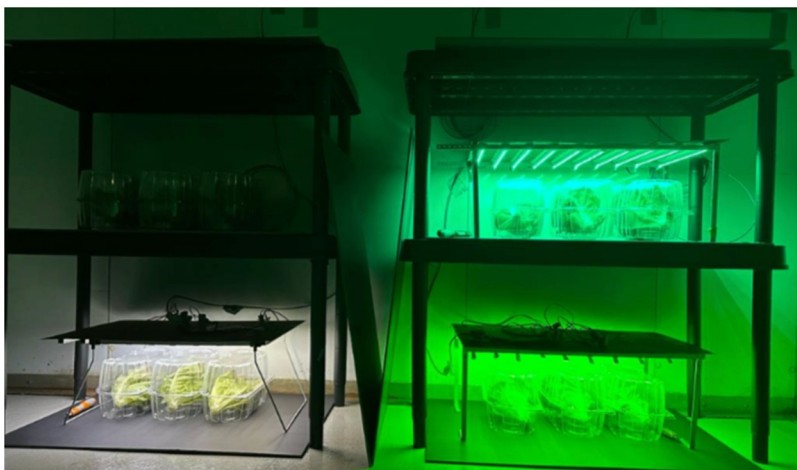

**Fig 7. Representative image of lettuces under different cold storage conditions, including dark storage (top left shelf), green LEDs with peaks at 500 nm (top right shelf) and 530 nm (bottom right shelf), and white LEDs (400–700 nm; bottom left shelf).**

(Model MQ500, Apogee Instruments Inc., Logan, UT, US). A consistent 12-h photoperiod was maintained to simulate the lighting conditions in retail stores.

## Packaging and storage conditions

Lettuce heads were randomly divided into four experimental groups, each consisting of three lettuces. Each individual lettuce head was placed within a transparent clamshell polyethylene terephthalate (PET) container (LBM AGTECH, Laval, QC, CA) measuring 30 cm (l) x 13 cm (w) x 15 cm (h). These groups were allocated to different shelves (dimensions: length 66 cm, width 32 cm, and height 27 cm), each outfitted with LEDs emitting distinct wavelengths (Fig 7). The shelves were shielded with black plastic to prevent light transmission to the other samples. The entire setup was situated in a cold storage room and maintained for a period of 14 d. A control group was included in the experiment and subjected to dark storage conditions. Throughout the duration of the experiment, the cold storage room was maintained at 5°C ±1°C and a relative humidity (RH) ranging from 90% to 95%. Temperature and humidity data were collected at five-minute intervals using a data logger (Elitech RC51H, Elitech Technology, San Jose, CA, US) installed within each shelf. The entire experiment was conducted in triplicate.

## Extract preparation

Three lettuce head samples were collected on d 0 (initial day) and freeze-dried using a laboratory-scale vacuum freeze-drier (Martin Christ Gefriertrocknungsanlagen GmbH Gamma 1–16 LSCplus, Osterode, Lower Saxony, Germany). Freeze-drying was carried out at room temperature for 48 h. After 14 d in cold storage, representative lettuce samples from each experimental light group and dark storage were collected. All freeze-dried samples were subsequently stored in a plastic bag in a dark space at room temperature (20°C). For extract preparation, 1 g freeze-dried lettuce sample was immersed in liquid nitrogen before grinding with a mortar and pestle. After 5 min (when there was < 1 mg change in mass), samples were transferred to 50-ml Falcon tubes and 30 mL absolute methanol (Thermo Fisher Scientific, Waltham, MA, US) was added. Falcon tubes were vortexed (Thermo Scientific vortex, Waltham,

MA, US) for 20 min at 40 g. The extracts were incubated in the freezer at -20˚C in the dark for 48 h. After 2 d, each sample was vacuum-filtered using Whatman TM filter paper (Thermo Fisher Scientific, Waltham, MA, US). The extracts were further diluted with methanol (1:6 ratio or 2 mL extract+10 mL methanol) and subjected to analysis.

## Total chlorophyl and carotenoid content

The lettuce extracts subjected to analysis using a UV/visible spectrophotometer (Ultrospec 2100 pro, Biochrom Limited, Cambridge, England), at wavelengths of 470 nm, 652.4 nm, and 665.2 nm. Total chlorophyll a (Chl a), chlorophyll b (Chl b), and carotenoid content were determined with Eqs 1, 2 and 3 below, outlined previously by Lichtenthaler and Buschmann [66]:

$$\text{Chl a} = 16.72 * \text{Abs}_{665.2} - 9.16 * \text{Abs}_{652.4} \tag{1}$$

$$\text{Chl b} = 34.09 * \text{Abs}_{652.4} - 15.28 * \text{Abs}_{665.2} \tag{2}$$

$$\text{Carotenoid} = (1000 * \text{Abs}_{470} - 1.63 * \text{Chl}_a - 104.96 * \text{Chl}_b) \tag{3}$$

Results are expressed on a dry mass basis. All tests and analyses were conducted in three replicates.

## Total anthocyanin content

Total anthocyanin content in the lettuce samples was determined using a pH differential method [70]. Briefly, a pH 1.0 buffer reagent was prepared by mixing 980 mL distilled water with 1.86 g KCl in a 1-L beaker. HCl was then used to modify the pH to 1.0 ± 0.05. The volume was brought up to 1L with distilled water in a 1-L volumetric flask. A reagent with a pH of 4.5 was prepared in a beaker by combining 54.43 g sodium acetate trihydrate ($CH_3CO_2Na \cdot 3H_2O$) with distilled water to a volume of 960 mL. Following the addition of HCl to modify the pH to 4.5 ± 0.05, the buffer was brought up to 1 L with distilled water in another 1-L volumetric flask. The lettuce extracts were diluted by combining the 2 mL extract with 10 mL of absolute methanol. Subsequently, two aliquots were prepared for each diluted extract: one containing 1 mL of diluted extract in 4 mL of a pH 1.0 buffer solution, and the other containing 1 mL of diluted extract in 4 mL of a pH 4.5 buffer solution. For the determination of the total anthocyanin content, the absorbance of the test portion diluted with pH 1.0 and pH 4.5 buffers was measured at 510 and 700 nm, respectively, using a plate reader (BioTek Synergy HTX Multimode Reader, Agilent Technologies, California, USA) and Eq 4:

$$\text{Anthocyanin pigment content} = (A * MW * DF * 1000)/\varepsilon \tag{4}$$

Where,

$$A = (\text{Abs}510 - \text{Abs}700)\text{pH } 1 - (\text{Abs}510 - \text{Abs}700)\text{pH } 4.5$$

MW (cyanidin– 3 –glucoside) = 449.2, DF = dilution factor, $\varepsilon$ (Molar absorptivity) = 26, 900. Results were expressed as mg of cyanidin-3-glucoside equivalent per g of dry mass. All tests and analyses were conducted in three replicates.

## Total phenolic content

Total phenolic content (TPC) was assessed employing the Folin-Ciocalteu reagent (FCR) protocol [71]. Initially, the FCR was diluted ten-fold with distilled water (1 mL FCR (Sigma-

Aldrich, Saint Louis, MI, US) and 9 mL distilled water). Next, 10 μL of each sample extract was combined with 75 μL diluted FCR solution, and this mixture was allowed to incubate at room temperature for 10 min. Following this incubation, 75 μL of 2% sodium carbonate ($Na_2CO_3$) solution (Sigma-Aldrich, Saint Louis, MI, US) was added, and the resulting mixture was vigorously vortexed for 15 seconds. It was left to incubate for 45 minutes in the absence of light. A blank solution consisting of 70% methanol was used for reference. The absorbance of each extract was then measured at a wavelength of 765 nm using a plate reader (BioTek Synergy HTX Multimode Reader, Agilent Technology, CA, US). To generate a standard curve, gallic acid was used at five different concentrations ranging from 0.01 to 0.08 g $L^{-1}$. The total phenolic content of the lettuce extracts was expressed as g of gallic acid equivalents per g of dry extract mass (g GAE $g^{-1}$). All tests and analyses were conducted in three replicates.

## Total flavonoid content

Total flavonoid content (TFC) of lettuce was quantified utilizing the aluminum chloride colorimetric method [71]. Specifically, 15 μL of each lettuce sample extract was combined with a mixture consisting of 45 μL of 70% methanol (Thermo Fisher Scientific, Waltham, MA, USA), 3 μL of 10% aluminum chloride hexahydrate (Sigma-Aldrich, Saint Louis, MI, USA), 3 μL of 1 M sodium acetate (Sigma-Aldrich, Saint Louis, MI, USA), and 84 μL of distilled water. Subsequently, this mixture was incubated for a duration of 40 minutes. Following incubation, the absorbance at a wavelength of 415 nm was recorded using a plate reader (BioTek Synergy HTX Multimode Reader, Agilent Technologies, California, US) and used for the determination of the total flavonoid content in duplicate measurements. To generate a standard curve, quercetin was used at concentrations ranging from 0.05 to 0.25 g $L^{-1}$. Total flavonoid content of the lettuce extract was expressed as g quercetin equivalents per g of dry mass (g QE $g^{-1}$).

## Antioxidant activity

The evaluation of antioxidant activity in lettuce was performed using the DPPH method as described previously [72] with some modifications. Briefly, 0.5 mL of diluted extract was introduced into a 1 mL methanolic solution containing DPPH radicals (0.2 mM). The mixture was vigorously vortexed (15 s) and allowed to stand for a duration of 30 minutes. The absorbance was measured at 515 nm using a plate reader (BioTek Synergy HTX Multimode Reader, Agilent Technologies, California, USA). Ascorbic acid (50mM) was employed at five different concentrations ranging from 0.01 to 0.32 g $L^{-1}$. All tests and analyses were conducted in three replicates.

## Total soluble solids

Total soluble solids (TSS) analyses of lettuce leaves were performed using a digital refractometer (ATAGO, model PR32a, Tokyo, Japan) as previously described [73]. A 5 g sample of lettuce leaves was completely ground, followed by centrifugation at 1372 g for 10 min to collect a liquid supernatant. A small quantity of this liquid was used for analysis, placed onto the inspection mirror of the digital refractometer. The refractometer's scale reading (%) was then recorded. All tests and analyses were conducted in three replicates.

## Statistical analysis

Statistical analyses were performed using SAS statistical analysis software developed by SAS Institute Inc. (Cary, NC, US). The experimental design employed a split-plot arrangement within a completely randomized design. The experiment repeated three times. The results

given in this study are shown as the mean ± standard deviation (SE). In order to ascertain significant disparities between means, Duncan tests, which were safeguarded by ANOVA, were utilized at a significance level of $p < 0.01$.

## Acknowledgments

The authors would like to extend their heartfelt appreciation to all their colleagues in the Biomass Production Laboratory for their unwavering support throughout the project, with a special mention to Mr. Yvan Gariepy for his invaluable assistance.

## Author Contributions

**Conceptualization:** Shafieh Salehinia, Mark Lefsrud.

**Formal analysis:** Shafieh Salehinia, Fardad Didaran.

**Funding acquisition:** Mark Lefsrud.

**Investigation:** Shafieh Salehinia.

**Methodology:** Shafieh Salehinia, Sasan Aliniaeifard, Mark Lefsrud.

**Supervision:** Mark Lefsrud.

**Writing – original draft:** Shafieh Salehinia.

**Writing – review & editing:** Sasan Aliniaeifard, Saman Zohrabi, Sarah MacPherson, Mark Lefsrud.

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
