## [Decision Letter · Decision Letter 0]

24 Jul 2024

PONE-D-24-22256Green light enhances the nutritional value and pigment preservation of lettuce during postharvest cold storagePLOS ONE

Dear Dr. Lefsrud,

Thank you for submitting your manuscript to PLOS ONE. After careful consideration, we feel that it has merit but does not fully meet PLOS ONE’s publication criteria as it currently stands. Therefore, we invite you to submit a revised version of the manuscript that addresses the points raised during the review process.

We look forward to receiving your revised manuscript.

Kind regards,

Sajid Ali

Academic Editor

PLOS ONE

Journal Requirements:

3. Thank you for stating the following financial disclosure: "This work was supported by UTechnology Corporation (Calgary, Alberta, Canada) and Natural Sciences and Engineering Research Council of Canada (NSERC #CRDPJ 524170-18)." 

Reviewers' comments:

Reviewer's Responses to Questions

**Comments to the Author**

1. Is the manuscript technically sound, and do the data support the conclusions?

Reviewer #1: Partly

Reviewer #2: Partly

Reviewer #3: Yes

2. Has the statistical analysis been performed appropriately and rigorously? 

Reviewer #1: Yes

Reviewer #2: Yes

Reviewer #3: Yes

3. Have the authors made all data underlying the findings in their manuscript fully available?

Reviewer #1: Yes

Reviewer #2: Yes

Reviewer #3: Yes

4. Is the manuscript presented in an intelligible fashion and written in standard English?

Reviewer #1: Yes

Reviewer #2: Yes

Reviewer #3: Yes

5. Review Comments to the Author

**Reviewer #1: **The present experiment was carried out to assess the effects of Green and White LEDs on lettuce. I have some concern:-

There are already available literatures on effects of different lights on postharvest physiology of the fresh produce. In the present study physiological aspects such as respiration, weight loss, decay etc. are missing. Enzymology also did not include that is important part of postharvest physiology.

Why only Green and white lights be chosen. I think inclusion of blue and red LEDs may give more concrete results.

As the photographs show that author might have made some kind of chamber for light treatments, but chamber has been remained open from two sides. How did author create controlled condition for uniform distribution of lights. Introduction part is poorly described and needs to be improved. I recommend a comprehensive revision and resubmit.

**Reviewer #2:** The manuscript is interested discussing about a non-chemical postharvest treatment. Some comments are here and in the attached reviewed manuscript need to be considered in order to improve the manuscript.

- The title should be specific for the nutritional quality.

- The abstract is informative and brief.

- the results are well-presented, however, not presented the full picture about the lettuce quality:

the plant weight loss, the decay, the microbiology, the pink discoloration of lettuce leaf

- missing the sensory analysis since the treatment is increasing the phenolics and the chlorophyll contents.

**Reviewer #3:** In the manuscript PONE-D-24-22256, the authors investigated green light enhances the nutritional value and pigment preservation of lettuce during postharvest cold storage. It is my opinion that there is some useful information in the manuscript. However, the following points may be considered for improving the quality of this manuscript.

Abstract

There is a mix of precise values and percentages, which can be confusing. Sticking to one form of data presentation might improve readability. The term "photoperiod (12 h-d) is unconventional and could be simplified to "photoperiod (12 hours per day).

Introduction

There is some repetition, particularly around the nutritional benefits of lettuce and the challenges of postharvest storage. Condensing these sections could make the introduction more succinct. The introduction mentions various methods for preserving postharvest quality (e.g., chemical treatments, edible coatings), but it could benefit from a clearer focus on why LEDs, particularly green LEDs, were chosen for this study.

While the introduction cites many studies, it could better synthesize this information to highlight gaps in the current research. For instance, summarizing key findings from previous studies on LED effects on vegetables and then pointing out what remains unknown about green LEDs specifically. The introduction mentions the economic importance of lettuce but doesn't link this directly to the benefits of improved postharvest storage. Making this connection explicit would strengthen the introduction.

What specific mechanisms are proposed for the effects of green LED light on bioactive compounds? More detail on the proposed biological processes would be useful.

Results

The initial values of total phenolic content are not provided, making it difficult to understand the magnitude of changes. Including these initial values would give a better context. There is a significant decrease in anthocyanin content for all treatments. However, there is a lack of explanation for why the 530 nm green LED significantly prevented this decrease while 500 nm did not. The percentage increases and decreases are clearly reported, but the initial values again are missing, which makes it harder to interpret the results fully.

Discussion

The PCA and correlation matrix results are presented with a focus on clustering patterns, but the interpretation of these patterns is minimal. Explaining why certain clusters form and what this indicates about the biochemical changes in the lettuce would be beneficial.

6. PLOS authors have the option to publish the peer review history of their article (what does this mean?). If published, this will include your full peer review and any attached files.

Reviewer #1: **Yes: **Dr. Nirmal Kumar Meena

Reviewer #2: No

Reviewer #3: **Yes: **Ghulam Khaliq

---

## [Author Response · Author response to Decision Letter 0]

7 Sep 2024

Editor:

Response:

Thank you for your comment. The manuscript title page and file naming have been revised to ensure that it meets PLOS ONE's style requirements.

2. PLOS requires an ORCID iD for the corresponding author in Editorial Manager on papers submitted after December 6th, 2016. Please ensure that you have an ORCID iD and that it is validated in Editorial Manager.

Response:

Thank you for the information. The ORCID iD has been linked and validated in Editorial Manager by following the required steps.

3. Thank you for stating the following financial disclosure: "This work was supported by UTechnology Corporation (Calgary, Alberta, Canada) and Natural Sciences and Engineering Research Council of Canada (NSERC #CRDPJ 524170-18)." Please state what role the funders took in the study. If the funders had no role, please state: ""The funders had no role in study design, data collection and analysis, decision to publish, or preparation of the manuscript."" 

Response:

Thank you for the information. The financial disclosure has been revised.

The statement reads:

This work was funded by UTechnology Corporation (Calgary, Alberta, Canada) and Natural Sciences and Engineering Research Council of Canada (NSERC #CRDPJ 524170-18). However, the funders had no role in study design, data collection and analysis, decision to publish, or preparation of the manuscript.

4. We note that you have indicated that there are restrictions to data sharing for this study. PLOS only allows data to be available upon request if there are legal or ethical restrictions on sharing data publicly. If there are no restrictions, please upload the minimal anonymized data set necessary to replicate your study findings to a stable, public repository and provide us with the relevant URLs, DOIs, or accession numbers. 

Response:

Thank you for the comment. We uploaded our data in dara repository. Here are the doi and URL:

doi:10.5061/dryad.vx0k6dk1s

URL:https://datadryad.org/stash/share/ngsVhWA13Txrbll92ZsrdhFgkmpA7sqCmyLnHTSUflk

5-We note that you have included the phrase “data not shown” in your manuscript. Unfortunately, this does not meet our data sharing requirements. PLOS does not permit references to inaccessible data. We require that authors provide all relevant data within the paper, Supporting Information files, or in an acceptable, public repository. Please add a citation to support this phrase or upload the data that corresponds with these findings to a stable repository (such as Figshare or Dryad) and provide and URLs, DOIs, or accession numbers that may be used to access these data. Or, if the data are not a core part of the research being presented in your study, we ask that you remove the phrase that refers to these data.

Response:

Thank you for your comment. The phrase ‘ data not shown’ has been removed from the manuscript and the statement has been cited to our previous study.

Statement reads (line 876-878) in revised manuscript with track change): 

The light intensity employed in this experiment was set at 10 µmol m-2 s-1 PPFD, based on our preliminary research [34]. 

6- While revising your submission, please upload your figure files to the Preflight Analysis and Conversion Engine (PACE) digital diagnostic tool, https://pacev2.apexcovantage.com/. PACE helps ensure that figures meet PLOS requirements.

Response: 

Thank you for your comment. All figure files have been uploaded to the PACE to ensure they meet PLOS requirements.

Reviewer #1: The present experiment was carried out to assess the effects of Green and White LEDs on lettuce. I have some concern:-

There are already available literatures on effects of different lights on postharvest physiology of the fresh produce. 

1. In the present study physiological aspects such as respiration, weight loss, decay etc. are missing. Enzymology also did not include that is important part of postharvest physiology.

Response

Thank you for the comment. We agree that including physiological aspects such as respiration, moisture loss, and decay in postharvest physiology are needed for a comprehensive understanding of postharvest quality. They have been addressed in a separate manuscript that is part of a parallel study. We are also currently conducting further experiments to assess enzymological factors, which will be detailed in a follow-up publication. This decision allows us to focus each paper on specific aspects of the research while ensuring that physiological, enzymological, and nutritional factors are adequately explored. We will reference this manuscript in our upcoming study to ensure that readers are aware of the broader scope of our research. 

The statement below has been added to the conclusion to emphasize the importance of measuring other factors in future studies.

The statement reads (lines 821-825) in revised manuscript with track change:

Further investigation into how storage under green LED in the cold storage affects quality parameters such as moisture loss, sensory analysis, respiration rate, enzymology and microbiology in leafy greens is warranted, and additional work will explore different combinations of green light with modified atmosphere packaging.

2. Why only Green and white lights be chosen. I think inclusion of blue and red LEDs may give more concrete results.

Response

Thank you for the comment. In our previous studies, we demonstrated that green light has a unique ability to induce stomatal closure compared to red and blue light, which can significantly reduce transpiration and moisture loss in plants (Rufyikiri et al., 2024; Rufyikiri, 2018; Salehinia, 2023). Based on these findings, we selected green LEDs in this current work to gather more detailed data on how this specific light spectrum affects the nutritional quality of lettuce during postharvest storage. Limited studies have explored the effects of green LEDs on bioactive compounds such as flavonoids and anthocyanins and this study will add to this body of literature. 

The introduction has been revised to address the need for explanation of the reason for choosing the green LED.

The statement reads (line 113-116) in revised manuscript with track change:

LEDs can positively influence the nutritional attributes of various fruits and vegetables [23-26], with most research focusing on red and blue light. However, limited studies have explored the effects of green LEDs on bioactive compounds such as flavonoids and anthocyanins, particularly in leafy greens.

The statement reads (line 148-158) in revised manuscript with track change:

Green wavelengths, specifically 500-560 nm, can induce stomatal closure more effectively than red and blue light, significantly reducing transpiration and moisture loss in spinach, kale, basil, and lettuce plants, thereby maintaining better visual quality [33, 34]. Despite these promising results, there is still limited understanding of which specific green spectra are most effective in preserving or enhancing the health-promoting properties of postharvest vegetables, especially anthocyanin and flavonoid. The aim of this work was to build on this foundational research by assessing the impact of two monochromatic green LED lights (500 nm and 530 nm) in comparison to white LED lighting (400-700 nm) and dark storage on the nutritional quality of lettuce heads stored at 5°C for 14 d.

3. As the photographs show that author might have made some kind of chamber for light treatments, but chamber has been remained open from two sides. How did author create controlled condition for uniform distribution of lights. 

Response

Thank you for your comment. The entire setup (the chamber) was housed within a cold storage room with controlled conditions. The chamber was constructed with openings on both sides to allow for adequate air circulation and ensure that the environmental conditions, such as temperature and humidity, remained consistent across all shelves during the experiment. The open sides helped prevent the buildup of heat or moisture within the chamber, which could affect the uniformity of light distribution and potentially introduce variables that confound our results. We ensured that all other factors were controlled and consistent across treatments to focus solely on the impact of the light spectra under study.

4. Introduction part is poorly described and needs to be improved. I recommend a comprehensive revision and resubmit.

Thank you for your constructive criticism. The introduction has been completely revised and improved.

The revised Introduction is as follows: 

Lettuce (Lactuca sativa) is a globally distributed and consumed leafy green, valued for its high water content, low-calorie count, and rich nutritional profile, including dietary fiber, minerals, vitamins, antioxidants, and phytochemicals such as carotenoids and phenolic compounds [1-4]. Maintaining lettuce quality postharvest presents substantial challenges due to its continued metabolic activity after harvest, leading to chlorophyll degradation, wilting, and decay [5-8]. These changes not only diminish the vegetable's appearance and nutritional value but lead to consumer rejection [1, 5-7]. Improving postharvest storage conditions directly benefits the economic viability of lettuce by extending its shelf life, maintaining its nutritional and visual quality for longer periods, and reducing waste. This ensures that more of the harvested crop reaches consumers in optimal condition, thereby maximizing market value and profitability.

Various methods have been employed to preserve postharvest quality, including chemical treatments, edible coatings, and modified atmosphere packaging [9-13]. While these techniques can be effective, they often come with concerns related to cost, effectiveness, practicality, and food safety [14, 15]. There is a growing interest in developing alternative methods that are both cost-effective and environmentally friendly. Among these, light-emitting diodes (LEDs) have emerged as a promising technology [16-19]. LEDs offer advantages such as precise control over light intensity and spectrum, non-toxicity, and cost efficiency[20-22].

LEDs can positively influence the nutritional attributes of various fruits and vegetables [23-26], with most research focusing on red and blue light. However, limited studies have explored the effects of green LEDs on bioactive compounds such as flavonoids and anthocyanins, particularly in leafy greens. Green LED light has been shown to influence various physiological and biochemical processes in plants, primarily through its interaction with specific photoreceptors like cryptochromes and phytochromes, which regulate numerous metabolic pathways. Unlike red and blue light, which predominantly activate photoreceptors at the surface level, green light penetrates deeper into plant tissues, thereby affecting internal photoreceptors [27, 28].

 Guo, Zhou [29] demonstrated that green LED lighting at 25 μmol·m²·s⁻¹ for 12 d inhibited malondialdehyde accumulation and maintained higher chlorophyll content in cabbage (Brassica oleracea var. capitata) by regulating glucosinolate metabolism, while Jin, Yao [30] found that broccoli florets (Brassica oleracea . ssp. italica) exposed to green LED light had higher sulforaphane content compared to those stored in darkness or fluorescent lighting for 2 d at 25°C. Castillejo, Martínez-Zamora [27] found that green LED increased total glucosinolate content by 34.5% although green LEDs had no impact on the total phenolic content on ready-to-eat broccoli sprouts (Brassica oleracea var. italica) during 15 days at 5 °C compared to traditional storage in darkness. These findings suggest that green LED light may influence biosynthetic pathways through pigment photoreceptor proteins, such as cryptochromes and phytochromes, leading to the accumulation of bioactive compounds, which are crucial for enhancing the nutritional quality of postharvest vegetables [27, 28].

Jin, Yao [30] demonstrated that green LED light (12 μmol m⁻² s⁻¹) increased total phenolic content and DPPH radical scavenging activity in broccoli florets. This effect was linked to enhanced activity of phenylalanine ammonia-lyase (PAL), a key enzyme in the phenylpropanoid pathway responsible for the synthesis of phenolic compounds. Green LED light has been shown to increase glucosinolate levels in broccoli, indicating that light quality and dosage can significantly influence phytochemical synthesis and overall nutritional quality [30].

 Pennisi, Orsini [31] found that green LED light helped maintain color and pigment content in rocket leaves (Diplotaxis tenuifolia) during storage, suggesting that light quality, particularly green light, is a critical exogenous factor in regulating senescence. The capacity of harvested leaves to respond to light stimuli, mediated by photoreceptors, underscores the importance of proper light management to balance functional metabolism, such as maintaining chlorophylls and carotenoids, with the natural progression of leaf senescence [31, 32].

Green wavelengths, specifically 500-560 nm, can induce stomatal closure more effectively than red and blue light, significantly reducing transpiration and moisture loss in spinach, kale, basil, and lettuce plants, thereby maintaining better visual quality [33, 34]. Despite these promising results, there is still limited understanding of which specific green spectra are most effective in preserving or enhancing the health-promoting properties of postharvest vegetables, especially anthocyanin and flavonoid. The aim of this work was to build on this foundational research by assessing the impact of two monochromatic green LED lights (500 nm and 530 nm) in comparison to white LED lighting (400-700 nm) and dark storage on the nutritional quality of lettuce heads stored at 5°C for 14 d. Antioxidant activity, anthocyanins, flavonoids, chlorophylls, carotenoids, and total soluble solids were measuered to better understand how these wavelengths influenced the nutritional quality of lettuce during postharvest storage. By improving postharvest storage conditions using green LEDs, it may be possible to reduce waste and provide consumers with higher quality produce.

Reviewer #2: The manuscript is interested discussing about a non-chemical postharvest treatment. Some comments are here and in the attached reviewed manuscript need to be considered in order to improve the manuscript.

1. Title should be specific for the nutritional quality.

Response

Thank you for the comment. The title has been revised. 

The title now reads:

Green light enhances the phytochemical preservation of lettuce during postharvest cold storage

2. - The results are well-presented, however, not presented the full picture about the lettuce quality:the plant weight loss, the decay, the microbiology, the pink discoloration of lettuce leaf. Missing the sensory analysis since the treatment is increasing the phenolics and the chlorophyll contents.

Response

Thank you for the comment. In this study, our primary focus was on the impact of green light on the nutritional quality of lettuce. we agree that other important aspects of lettuce quality, such as mass loss, decay, pink discoloration, and sensory analysis are essential for a better understanding of postharvest quality. We have addressed these in a separate manuscript that is part of a parallel study. This decision allows us to focus each paper on specific aspects of the research while ensuring that both physiological and nutritional factors are adequately explored. We will reference this upcoming manuscript in our current study to ensure that readers are aware of the broader scope of our research. This approach allows us to comprehensively evaluate lettuce quality, addressing the full range of factors you have highlighted.

The statement has been added to the conclusion to emphasize the importance of measuring other factors in future studies.

The state

---

## [Decision Letter · Decision Letter 1]

13 Sep 2024

Green light enhances the phytochemical preservation of lettuce during postharvest cold storage

PONE-D-24-22256R1

Dear Dr. Lefsrud,

We’re pleased to inform you that your manuscript has been judged scientifically suitable for publication and will be formally accepted for publication once it meets all outstanding technical requirements.

Kind regards,

**Sajid Ali**

Academic Editor

PLOS ONE

Additional Editor Comments (optional):

Reviewers' comments:

Reviewer's Responses to Questions

**Comments to the Author**

1. If the authors have adequately addressed your comments raised in a previous round of review and you feel that this manuscript is now acceptable for publication, you may indicate that here to bypass the “Comments to the Author” section, enter your conflict of interest statement in the “Confidential to Editor” section, and submit your "Accept" recommendation.

Reviewer #2: All comments have been addressed

Reviewer #3: All comments have been addressed

2. Is the manuscript technically sound, and do the data support the conclusions?

Reviewer #2: Yes

Reviewer #3: Yes

3. Has the statistical analysis been performed appropriately and rigorously? 

Reviewer #2: (No Response)

Reviewer #3: Yes

4. Have the authors made all data underlying the findings in their manuscript fully available?

Reviewer #2: Yes

Reviewer #3: Yes

5. Is the manuscript presented in an intelligible fashion and written in standard English?

Reviewer #2: Yes

Reviewer #3: Yes

6. Review Comments to the Author

Reviewer #2: The author responded to all comments and addressed them through out the manuscript. the response satisfying the reviewers.

Reviewer #3: After revision, the authors have adequately addressed most of the issues raised during the review process. The revised manuscript has significantly improved. With these revisions, I now consider the manuscript to be acceptable for publication. I recommend accepting the manuscript for publication in its current form.

7. PLOS authors have the option to publish the peer review history of their article (what does this mean?). If published, this will include your full peer review and any attached files.

Reviewer #2: No

Reviewer #3: **Yes: **Ghulam Khaliq

---

## [Editor Report · Acceptance letter]

19 Sep 2024

PONE-D-24-22256R1 

PLOS ONE

Dear Dr. Lefsrud, 

I'm pleased to inform you that your manuscript has been deemed suitable for publication in PLOS ONE. Congratulations! Your manuscript is now being handed over to our production team.

Kind regards, 

on behalf of

Dr. Sajid Ali 

Academic Editor

PLOS ONE